# Spatio-Temporal Extraction of Surface Waterbody and Its Response of Extreme Climate along the Upper Huaihe River

Hang Wang [1,2,*], Zhenzhen Liu [2], Jun Zhu [2], Danjie Chen [2] and Fen Qin [2,*]

1 Department of Geographic Sciences, Hanshan Normal University, Chaozhou 521041, China
2 Key Laboratory of Geospatial Technology for the Middle and Lower Yellow River Regions, Ministry of Education, Henan University, Kaifeng 475004, China; 18736958905@163.com (Z.L.); zhujun@vip.henu.edu.cn (J.Z.); chendanjie@henu.edu.cn (D.C.)
* Correspondence: wanghang20001@163.com (H.W.); qinfen@henu.edu.cn (F.Q.)

**Abstract:** The upper Huaihe River is the water-producing area of the Huaihe River Basin and the major grain and oil-producing area in China. The changing global climate over the recent years has increased the frequency of extreme weather in the upper reaches of the Huaihe River. Research on the responses of surface water bodies to extreme climates has become increasingly important. Based on all utilizable Landsat 4–8 T1–SR data and frequency mapping, the spatio-temporal extraction of surface water and its response to extreme climate were studied. We generated high-precision frequency maps of surface water, and a comparison of cartographic accuracy evaluation indices and spatial consistency was also carried out. The high-precision interpretation of small waterbodies constructs a surface water distribution with better continuity and integrity. Furthermore, we investigated the effect of El Niño/La Niña events on precipitation, temperature, and surface water along the upper Huaihe River, using the Mann–Kendall mutation tests. The results show: in 1987–2018, periods of abrupt changes in precipitation coincide with EI Niño/La Niña events, indicating that the precipitation was sensitive to EI Niño/La Niña events, which also strongly correlated with surface water area during wet and dry years. The effect of extreme events on seasonal water was smaller than permanent water. Surface water area showed an insignificant declining trend after 1999 and a significant drop in 2012. The phenomenon of topographic enhancement of precipitation controlled the spatial distribution of permanent water, with human activities having a substantial effect on the landscape pattern of seasonal water. Finally, discussions and applications related to the Markov Chain probability calculation theory in the paper contributed to enriching the theories on frequency mapping. The relevant results provide a theoretical basis and case support for the formulation of long-term water resources utilization and allocation policies.

**Keywords:** surface waterbody; time-series mapping; extreme climate; El Niño/La Niña

## 1. Introduction

Time-series data have clear advantages for monitoring changes in environmental characteristics [1]. Remote sensing cloud platform for remote sensing data processing and analysis on the large scale will be a key direction of development in the field of land cover mapping [2]. The mapping method considering the time dimension is obviously better than single-point classification [3]. Multi-seasonal imagery has been shown to improve the accuracy of forest biomass estimation [4]. Furthermore, the benefits of multi-temporal data for the estimation of successional processes have been acknowledged [5]. In time-series monitoring remote sensing mapping, the same sensor spectrum products are superior to multi-source image products in terms of spatial consistency. Landsat satellite images have been favored by scholars because of their long acquisition duration, abundant archived data, and high spatial resolution [6]. However, series satellite images have specific data missing and poor data quality [7], posing a challenge to the production of time series

remote sensing products. The pixel classification, using effective observing frequency to produce images brings opportunities for the time-series products. For a long-sequence Landsat image set, pixel-level synthetic images are generated by gathering a sufficient number of Landsat images to form a series of time-series pixels. The ability to synthesize the values of land cover attributes at one location over multiple time phases facilitates more accurate predictions, and then produces cloudless, seamless, and continuous synthetic images. Studies have used "per pixel frequency" to calculate the high variation of surface water in the recent past [8,9]. This method is very suitable, particularly for the regions with sparse cloud free data [8]. Due to the complexity of regional hydrodynamics, an accurate estimation of pixel attributes is critical to time-series mapping. Pixel frequency calculations need to be discussed theoretically and practically. By exploring the probability theory of Markov chain, this study discusses the frequency statistical method of "water" and "non-water" classification on the interannual scale and improves the theory of pixel frequency calculation.

Surface water is the ecosystem most sensitive to climate change. As its composition, structure, distribution, and function are all related to climatic factors, significant changes in the climate will undoubtedly have a substantial impact on wetland ecosystems [10]. The upper reaches of the Huaihe River are located in the transition zone between the subtropical humid region and the temperate sub-humid region. Affected by the East Asia monsoon, the region shows significant interannual variation in precipitation, a large rate of change of precipitation in the main flood period, and frequent droughts and floods. To control flood and drought, several hundred large and medium-sized reservoirs, ponds and lakes have been built, showing significant water environment effects from the artificial seasonal water bodies. After 1991, frequent cases of "abrupt alteration from drought to flood" have been observed in the upper reaches of the Huaihe River, especially after 2000 [11]. In such a region with a significant extreme climate, studies on an individual water body can no longer meet the needs of basin-wide drought and flood control. Greater attention should be given to prevalent small river networks and other seasonal water bodies. Image extraction and mapping for small water bodies, especially automatic time-series mapping, is the key to researching the spatio-temporal changes of seasonal water bodies.

On top of the above points, this paper intended to harness the power of the GEE remote sensing cloud platform and all utilizable Landsat 4-8 T1-SR data, to construct a water body extraction model and a frequency mapping model, realizing time-series mapping of surface water in the upper reaches of the Huaihe River with a high degree of automation. Based on this, precipitation, temperature, and other climate data were incorporated to study the response relationships of surface water to extreme climate. Discussions and applications related to the Markov Chain probability calculation theory in the paper will contribute to enriching the existing theories on frequency mapping. The high-precision image extraction and mapping of surface water in the upper reaches of Huaihe River, as well as the research on their responses to extreme climates, will also provide theoretical supports to water resource management in the Huaihe River Basin, especially water resource governance under extreme weathers.

## 2. Materials and Methods

### 2.1. Study Area

The Huaihe River is one of China's seven major rivers. It is a perennial river that originated from Tongbai Mountains in Tongbai County, Nanyang City, Henan Province. The curvature of Huaihe River is 2.35, its main stream runs through Henan, Anhui and Jiangsu and flows into the sea in Yangzhou and Yancheng of Jiangsu Province. The study area was located in the upstream area of Huaihe River, Henan Province, China, covering an area of 36,000 km$^2$, located in 113–116° E and 31–33° N. In the south, Tongbai Mountain and Dabie Mountain stand from west to east. The entire region is predominantly occupied by mountains and hills. Moderately high mountains over 1000 m in elevation cover approx. 2000 km$^2$ and low mountains and hills below 1000 m cover around 30,000 km$^2$. The altitude

for the area north of the mountains in southern Henan is only 50–100 m (Figure 1b). The research area shows a gradient reduction in elevation from the west and south to the east and north, in a gradually widening horseshoe shape [12]. Under the influence of the monsoon climate, the water system in the upper reaches of the Huaihe River shows clear changes in runoff volume. Precipitation concentrated in flood season, and great changes in interannual rainfall.

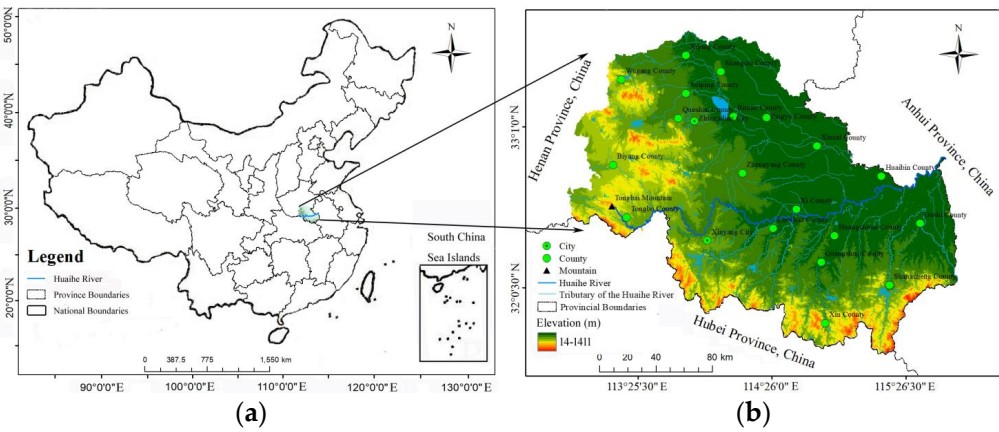

**(a)**                    **(b)**

**Figure 1.** Location map of the study area, (**a**) Location of the study area within China with province boundaries, (**b**) Location of the study area within Henan Province.

The study area is an important production base of grain, cotton, and oil plants in China, and one of the regions in China with the most frequent human activities down the history. The study area spans across 21 districts, counties, and cities in Henan Province. As of the end of 2019, the permanent resident population in the covered area registered about 13.9 million, with a GDP of CNY 598.813 billion. Along with the population increase and economic and social development, under the coupling effect of natural and human factors, the study area has suffered from frequent floods, droughts and storm surges [13]. Statistics show that the frequency of flood and drought in these areas is twice every three years and once every two years, respectively. The number of disaster-stricken years accounts for over 90% of the entire statistical timeline. Quite a few years were hit by both floods and droughts, and successive years of floods and droughts were also commonly seen [14]. In light of the frequent droughts and floods, the Huaihe River became the first river to be improved in a comprehensive and planned manner after the founding of the People's Republic of China.

In this paper, refereeing to the commonly used classification of seasonal and perennial rivers, surface water was divided into seasonal and permanent water by considering the temporal variation of surface water in the study area. Seasonal water included river and lake beaches, tributary water systems, irrigation channels, ponds (simple dams, common in rural fields and villages and constructed by hand for storing rainwater or small amounts of mountain water, gully water, and similar water bodies, used for agricultural irrigation, breeding, and domestic uses). Permanent water included areas covered perennially by water, including reservoirs, perennial rivers, and lakes.

## 2.2. Collection and Preparation of Data

### 2.2.1. Landsat Images

We used the following datasets: T1-SR TM, ETM+,Landsat 4\5\7 T1–SR TM, ETM+, and Landsat 8 Operational Land Imager (OLI). The period of study was January 1987 to 31 December 2018. All data were obtained from data sets on GEE platform, and data preprocessing and water extraction mapping were carried out using online computing methods. We conducted a human–computer interactive interpretation by employing six scenes of Landsat 8 OLI images from 18 October 2015 to 3 December 2015, with 96% overall mapping accuracy. We considered this map as the surface water base map for validation of

water classification mapping in 2015. An accurate evaluation dataset was sourced from the European Commission Joint Research Centre (JRC, Ispra, Italy).

### 2.2.2. El Niño/La Niña Data

The drought and flood data come from the literature [13] and the Henan Statistical Yearbook [15]. We obtained El Niño/La Niña data and its Fdiscriminant method according to A discriminant method for El Niño/La Niña events [16].

### 2.2.3. Rainfall and Temperature Data

Rainfall and temperature data were sourced from the daily meteorological dataset of the China Meteorological Administration for the period spanning January 1987 to December 2015. Using the MySQL (MySQL AB, Cupertino, CA, USA) database platform and ArcGIS10.2 (Esri, Redlands, CA, USA) software, we implemented several tasks such as determining the annual mean values and conducting ordinary kriging interpolation. Furthermore, we calculated regional statistics using data from 53 meteorological stations in the study area and its surroundings to obtain information, such as the annual rainfall, annual maximum temperature, and annual average temperature of the study area and each sub-basin.

### *2.3. Methodology*
### 2.3.1. Waterbody Extraction Model

GEE is a cloud platform for computing satellite images and other Earth data. In the era of big data, GEE facilitates rapid calculation and analysis of multi-source data. First, to collect and process images, we set the time nodes in GEE to year, month, and day. Processing included removing low frequency noise caused by clouds, cloud shadows, and mountain shadows, and filling in no-data regions. Second, refer to the approach of geographical zoning, three thematic indexes, which are as follows: the Modified Normalized Difference Water Index (MNDWI), Soil Adjusted Vegetation Index (SAVI), and Normalized Difference Building Index (NDBI), were used to construct two sets of water extraction models in the plain and mountainous areas, namely [(MNDWI $> -0.1$ or $AWEI_{sh} > -0.05$) and (SAVI $< 0.6$) and (NDBI $> -0.6$)]. In combination with Otsu's thresholding method, we extracted surface water data and generated an interannual water frequency map. We calculated the dynamic variable of surface water using the method proposed by [17] and the Mann–Kendall model proposed by the authors of [18]. The advantages of the Mann–Kendall trend test include less impact by the outliers and no requirements on distribution patterns. In addition, it is suitable for hydrology, meteorology, and other fields.

### 2.3.2. Frequency Calculation Model

Frequency Ratio (FR) is a univariate probability analysis method based on determining whether a pixel is water or non-water. The FR between the number of observations of the water pixels divided by the effective observation times is used for mapping. The frequency ratio model was successfully used in the study of the susceptibility of flood and insecurity in various flood-prone regions worldwide [19]. FR is a suitable method, but few theoretical models and processes of surface water frequency mapping have been constructed, which is not conducive to the popularization and comparative application of frequency mapping technique. In this paper, the probability calculation theory of Markov chain was used for discussion and application.

In Bayesian networks, directed acyclic graphs are used to represent the relation between random variables [20]. The node of a directed acyclic graph represents a random variable, and when any two nodes are connected by a one-way arrow, it indicates a causal relationship between the two random variables. Figure 2a is a Markov chain. Given the dis-

tribution of "a" and the conditional probability of the subsequent nodes, its joint probability calculation diagram is shown in Figure 2b, and its calculation model is as follows:

$$P(a, b, c) = P(a)P(b|a)P(c|b) \tag{1}$$

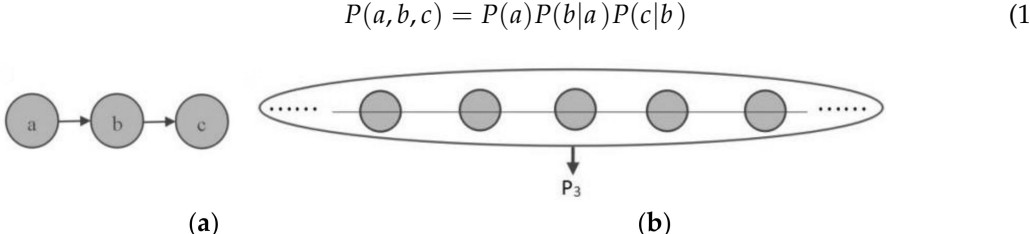

|     |     |
| --- | --- |
| (**a**) | (**b**) |

**Figure 2.** (**a**) Chain-structured Bayesian network; (**b**) Joint probability calculation of the time chain.

In the area where the two types of water meet, using probabilistic methods to distinguish them is also consistent with the objective fact that they coexist. Owing to regional stationarity [21] and time-series autocorrelation characteristics [22], on an interannual time scale, a time series of discretely distributed surface water pixels is similar to a Markov process and conforms to the phenomenon that the two types of water mainly depend on precipitation. In addition, using probability distribution and the transition probability statistical characteristics of the image source in information theory, [16,18] proved that the gray image could be used as a regionally stable Markov source. Meanwhile, the surface water frequency map is exactly a kind of gray image representing the general distribution of surface water. Therefore, it is theoretically feasible to use the Markov chain theory to effectively estimate the spatial distribution of surface water.

With the help of the FR mapping model, water is extracted by integrating all the "good" pixels of the year, and then used the Markov chain joint probability model to divide the attributes of the pixels. The process is highly automated, and the results are more accurate. The calculation formula of interannual water FR mapping is:

$$SWO = \sum WD.\text{year} / \sum AO.\text{year} \tag{2}$$

where surface water occurrence (*SWO*) represents the surface water frequency, and the range of values is FR $\in$ [0,1]. Water detection (*WD*) is the number of water observations and represents the number of images in which the pixel value is "1", and all observations (*AO*) is the number of valid observations and represents the number of effectively utilized images on pixels within the time node.

### 2.3.3. Water Classification Mapping

The acyclic digraph Markov model facilitates expressing the joint frequency of the whole time series directly, such that a cartographer can set the segmentation values of two ground features directly on an interannual scale to conduct time-series classification. Specific to the FR maps of interannual water, each pixel in an image is assigned a detailed FR value; however, the slight differences in FR values cannot be distinguished in this classification. We adopted the reclassification of the raster data technique to classify the FR value according to the order of magnitude; that is, the FR value within an order of magnitude was given the same grade, and finally, the classified product was used for classification evaluation. The reclassification process was realized by means of a histogram tool. Histogram gray-scale image segmentation is a simple and effective segmentation method, which can achieve superior segmentation for images with simple image content, and large grayscale differences between the target and background [23]. The histogram analysis method based on time-frequency information has a greater ability to suppress noise interference compared with other methods [24].

The higher the FR value in the FR maps, the greater the possibility of permanent water. This is reflected in the histogram as the phenomenon of high FR value and highly concentrated pixels. The classification breakpoint could be set to obtain an indication of permanent water. The FR value of seasonal water is low, but the number of pixels is large; therefore, the breakpoint could be set according to this phenomenon to obtain an indication of seasonal water. In order to avoid the fluctuations of the FR histogram in the 32 FR maps, we determined the approximate segmentation threshold by comparing the manual interpretation images in 2015, and then manually fine-tuned images in 32 FR maps to obtain more accurate surface water classification maps. For instance, when the breakpoint is $0.92 \leq FR \leq 1$, waterbodies such as large reservoir pits, main river streams, and large tributaries are extracted accurately; when $0.75 \leq FR < 0.92$, river and lake tidal flats are obtained; and when the breakpoint is divided by a threshold of $0.68 \leq FR < 0.75$, dotted ponds can be obtained. Figure 3 shows the technical details of this procedure.

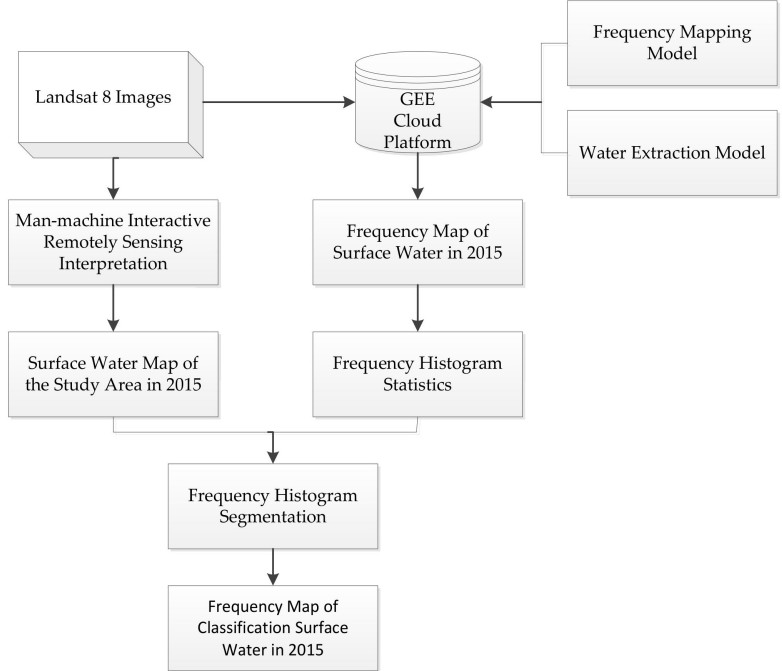

**Figure 3.** Technical route of water classification.

## 3. Results

### 3.1. Accuracy Evaluation

We obtained 32 FR maps of surface water, and sequentially calculated the overall accuracy, mapping accuracy, and Kappa coefficient [25]. The overall accuracy and Kappa coefficient were above 92% and 84% (Figure 4). Subsequently, we conducted the man–machine interactive water classification operation. We calculated the surface water areas for the period 1987–2018. In the past 32 years (Figure 5), the permanent water area ranged between 181.42 and 441.36 km$^2$, the standard deviation of the 32-year data was 67.68, and the coefficient of variation was 0.22. The tidal flat area varied between 180.86 and 550.90 km$^2$, the standard deviation was 106.37, and the variation coefficient was 0.30. The small pit area varied between 363.1 and 1357.22 km$^2$, the standard deviation was 277.64, and the variation coefficient was 0.39. According to the area variation coefficient of the three types of water, on the interannual time scale, the stability of the waterbody ranked in the following order: permanent water > tidal flat > ponds.

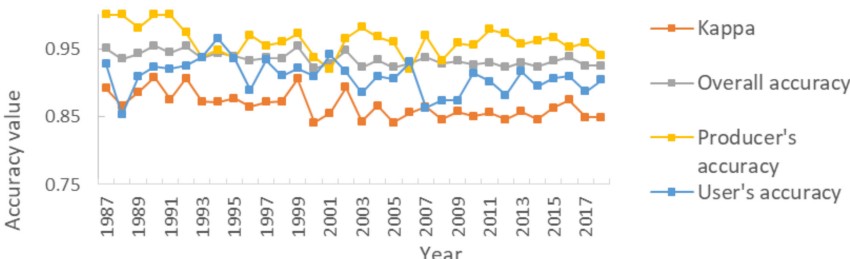

**Figure 4.** Accuracy evaluation of the water frequency chart from 1987 to 2018.

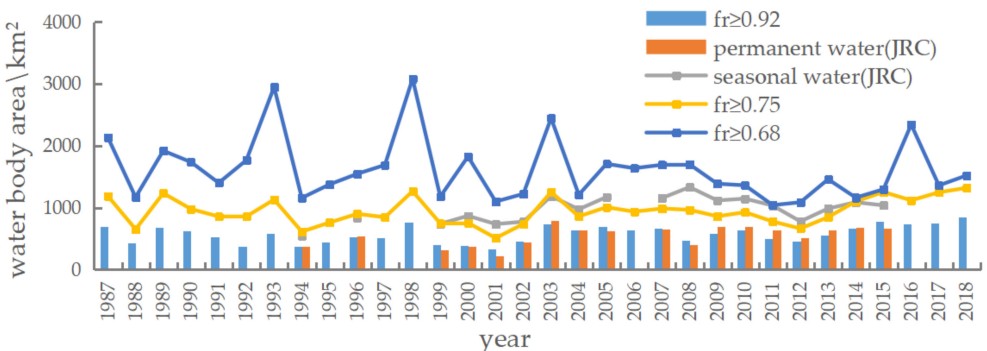

**Figure 5.** Area statistics of permanent and seasonal waterbodies in the study area.

We employed the JRC yearly historical dataset, which provides 32 periods of water detection from April 1984 to October 2015. Each detection was divided into four categories, namely permanent water, seasonal water, no data, and non-water. We divided the yearly historical data set into 32 periods of data instances, namely statistically permanent water, seasonal water, and non-water areas. In this area, a significant part of the JRC dataset is incomplete for the period between 1987 and 2015, with data lacking specifically for the years spanning 1984 to 1993, 1995 to 1998, and for 2006 (Figure 5). Upon analyzing the years with complete data, we found no significant difference between the two sets of data on permanent water, as indicated by the mean and standard deviations of the two datasets. The permanent water area range of the JRC is $257.78 \pm 29.45$ (array average $\pm$ standard deviation), and the permanent water area range of this study was $275.87 \pm 44.53$. In other words, the permanent water area of the JRC is slightly smaller than that indicated by this study.

Subsequently, we evaluated the space consistency of our FR maps. Figure 6a shows the 2015 surface water classification map obtained in this study, and Figure 6b shows the JRC 2015 surface water classification map. Figure 6a clearly shows most of the reservoir pits, main streams, and main tributaries within the FR > 0.92 range. The enlarged figure of the Xiaohong River system indicates that small rivers in the northeast plain of the study area in Figure 6a are absent in Figure 6b. Spatial continuity and location consistency are important evaluation indices to land cover products [26]. For the river network, both erroneous and missing extraction would cause a change in the river morphology. So, Figure 6a shows a surface river system that is more complete and in which the distribution is closer to the actual situation.

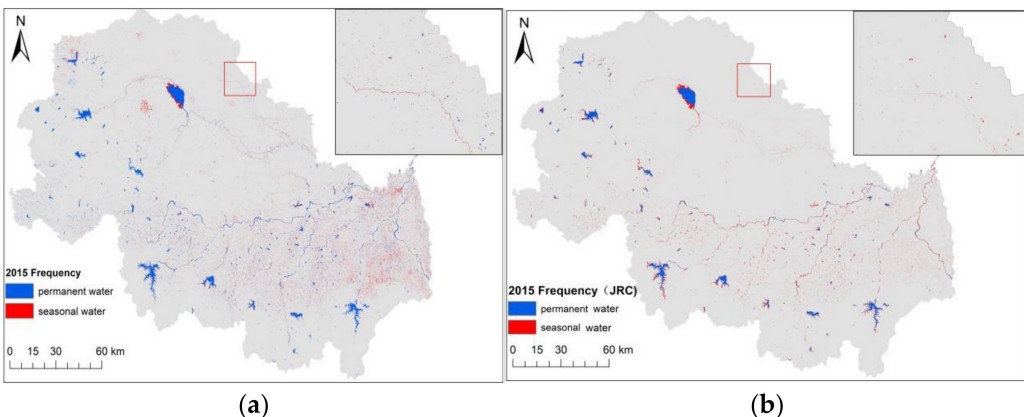

**Figure 6.** (**a**) Surface water classification maps from this study for 2015, and (**b**) surface water classification map of the European Commission Joint Research Centre (JRC) for 2015.

*3.2. Effects of Extreme Climate on Surface Water*

3.2.1. Extreme Precipitation, Drought, and El Niño/La Niña

Features of atmospheric circulation in typical flood/drought years indicate that the El Niño/La Niña are the main impact factors to flood/drought in the Huaihe Basin [13]. The current study sought to determine the response relationship between precipitation and the El Niño/La Niña phenomena from 1987 to 2015, and how such events cause changes in surface water. According to the discriminant method for El Niño/La Niña events [16], the type of events that are moderate or higher, as well as annual precipitation and annual surface water area, are integrated into map-making. Because of precipitation and temperature data were collected from 1987 to 2015, only the data of this period was discussed in the following chapters.

We calculated the high-precipitation level by dividing the total volume of rainfall in a high-precipitation year by the number of high-precipitation years, and the normal-precipitation level by dividing the total normal-precipitation year by the number of normal-precipitation years. Similarly, the total volume of precipitation in a dry year was divided by the number of dry years to calculate the low-precipitation level. As shown in Figure 7, there were 10 wet years, 3 abnormally wet years, 13 normal years, and 4 dry years from 1987 to 2015. This result equates to 43% normal years and 57% abnormally wet years, indicating that extreme weather occurred frequently in the study area. Based on the occurrence period and intensity peak of El Niño/La Niña, this weather system was extraordinarily intense in November 1997 and December 2015, and the annual precipitation in 1998 and 2015 reached the peak of that in the preceding three years. El Niño/La Niña events reached their peak at moderate intensity in December 1994, November 2002, and December 2009, with precipitation peaking in 1995, 2003, and 2010. In January 2000, during a La Niña event of moderate intensity, the precipitation peaked accordingly. Wet years are generally consistent with La Niña events; accordingly, 2007, 2008, and 2010 were indicated as wet years, whereas 2000 was an abnormally wet year. Clearly, El Niño/La Niña events have had a significant effect on precipitation in the study area. Concurrent with an El Niño event, cold air moves southward in early summer. At this time, the western Pacific Ocean current area is an evident anti-cyclonic anomaly, leading the warm and wet air flow northward. Consequently, the cold and warm air flow converges over the Huaihe Basin, resulting in more precipitation during the flood season. Similarly, the warm and wet airflow is transported to the Jianghuai area during a La Niña development year, which is conducive to heavy precipitation in this area. These findings are consistent with those of Luo et al. [13] and with the occurrence of high-precipitation and low-precipitation periods in the study area. However, not all El Niño/La Niña events cause heavy rainfall. For example, in 1988, 1994, and 1995, the precipitation was neither positively nor negatively correlated with El Niño/La Niña events [27–29]. By referring to the atmospheric circulation of the upstream

Huaihe River, it was the transportation of the southwest jet stream, leading to these heavy precipitation processes [30–33].

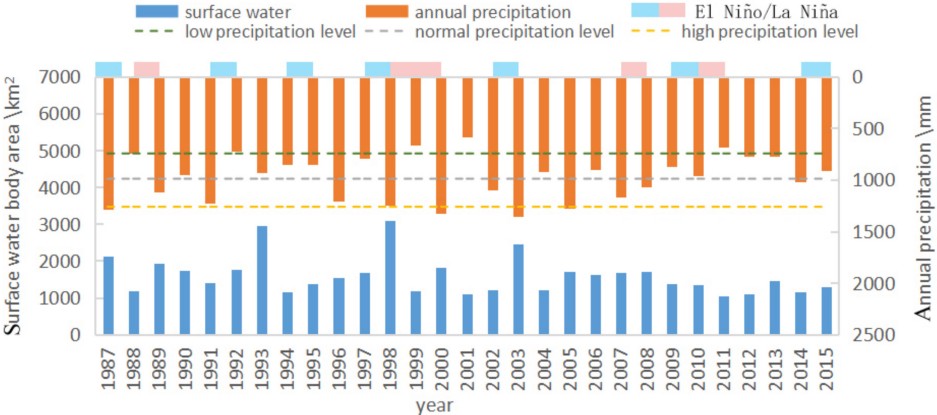

**Figure 7.** Precipitation, surface water area, and El Niño/La Niña events in the study area.

### 3.2.2. Time-Series Analysis of Climatic Factors and Surface Water

We used the Mann–Kendall method to test the time series of temperature, precipitation, and the two types of surface water. Figure 8b shows eight abrupt changes in rainfall occurring in 1987, 1989, 1990, 1995, 1997, 1999, 2008, and 2014, consistent with the occurrence of El Niño/La Niña events. In particular, during the period of high incidences of El Niño/La Niña events from 1987 to 1999, the precipitation changed several times, verifying the strong response relationship between precipitation and El Niño/La Niña events. Annual precipitation has increased since 1999, with a significant increase between 2003 and 2010, showing a downward trend after 2011 and starting to increase again after 2014. It can be seen from Figure 8c that the permanent waterbody showed periodic oscillatory changes before 2004 and sudden changes in 2004, 2008, and 2014; it decreased significantly in 1993 and 2001 and, later, showed a linear upward trend after 2005. Figure 8d shows that the seasonal water area has been on a downward trend, with a sudden decrease in 2009 and a significant decrease since 2011. The change in surface water area is not only affected by precipitation but is also indirectly affected by air temperature. Figure 8a shows that since 1993, the temperature in the study area has shown an upward trend but has increased significantly since 1999. Given the evident influence of global warming, rising temperatures, and increased evapotranspiration, the extensive areal distribution of seasonal waterbodies has been decreasing since 1993.

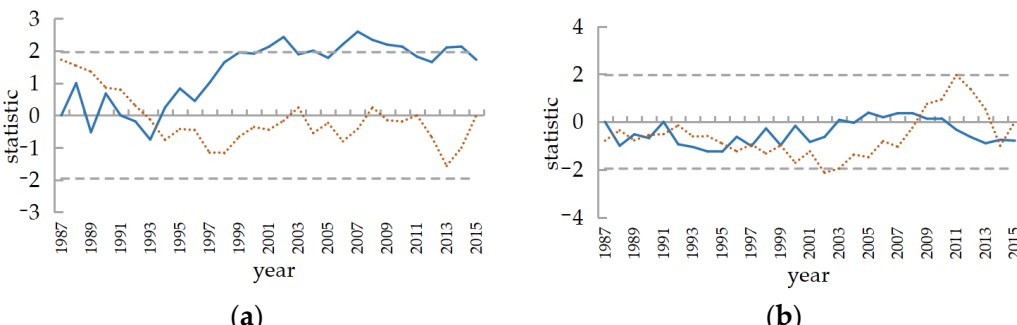

(**a**)          (**b**)

**Figure 8.** *Cont*.

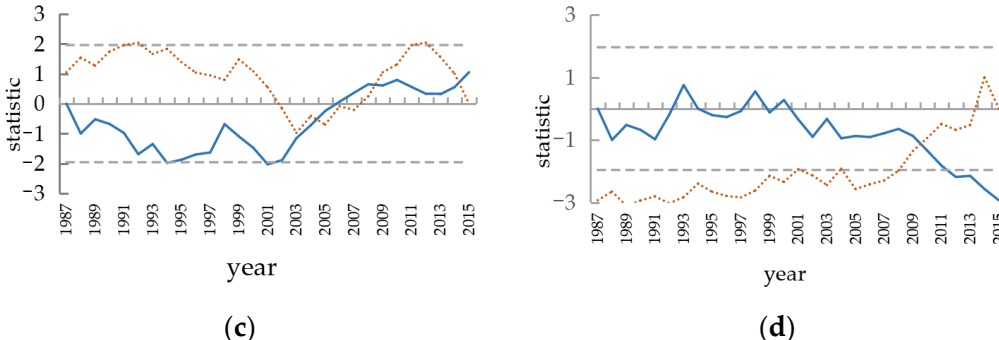

**Figure 8.** Abrupt change test of annual sequences of air temperature, precipitation, and surface water.
(**a**) Annual mean temperature, (**b**) annual precipitation, (**c**) permanent water area, and (**d**) seasonal
water area (dotted line indicates a significance level of 0.05).

To further analyze the influence of precipitation and temperature on surface water-bodies, we divided the study area into 31 sub-catchments by employing the hydrological analysis module of the ArcGIS 10.2 platform (Esri, Redlands, CA, USA). We used the Pearson's correlation coefficient to calculate the correlation among annual precipitation, annual average temperature, seasonal water area, and permanent water area. Figure 9 shows that on the sub-basin scale, correlation of the two types of surface water to precipitation is significantly higher than that of the two types of surface water to temperature. The correlation between the two types of climatic factors and seasonal water is higher than between climatic factors and permanent water. Among these results, the correlation coefficients between precipitation and seasonal water essentially passed the 0.05 significance test. We observed correlation coefficient curves of the precipitation and the two types of surface water, with the two curves showing a consistent change trend and correlation that is more significant in wet years than in dry years. The correlation coefficient curve between temperature and the two types of surface water indicated that correlation coefficient between temperature and permanent water decreased since 2002. We found essentially no correlation between the permanent water area and temperature in any sub-catchment since 2002. In contrast, the correlation between temperature and seasonal water has increased significantly since 1999 and particularly during 2007–2009. The correlation values between the maximum temperature and seasonal water were 0.47, 0.65, and 0.62 (i.e., above the level of significance), with the continuously high temperature leading to an increase in the loss of seasonal water. Furthermore, the hydrological effect of extreme precipitation is clearly evident (e.g., in 1987, 1988, 1993, and 1998), and the correlation between precipitation and the seasonal water area increased significantly. This result is ascribed to heavy precipitation in extreme years being accompanied by high temperatures (r = 0.77, $\alpha$ = 0.05). Owing to the extensive distribution of seasonal water and the need for ecological water on land, evapotranspiration increased, which led to the most sensitive seasonal water being affected to a greater extent. In addition, replenishment of stable water caused a time lag between permanent water and the maximum temperature (e.g., from 2009 to 2010); the correlation coefficients were 0.42 and 0.61, respectively, reaching significant correlation at the $\alpha$ = 0.01 level. It should be pointed out in particular that the pre-flood season regional drought in 2000, the phenomenon of "abrupt alteration from drought to flood" occurred before the plum rain season. Compared with other years with the phenomenon, precipitation in this year was greater and more concentrated. Following the alteration, there was basin-type flood. On the interannual scale, it showed more precipitation, but a smaller seasonal water body for extended dry period. There was a negative correlation between the precipitation and seasonal water area.

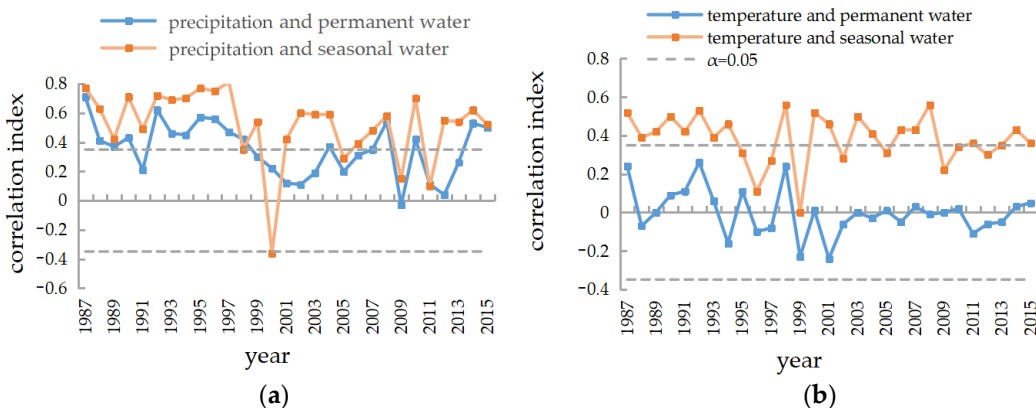

**Figure 9.** Interannual sequence correlation between rainfall, temperature and surface water. (**a**) precipitation and surface water, and (**b**) temperature and surface water.

### 3.2.3. Effect of Climate on the Distribution Pattern of Surface Water

Topography and geomorphology are the main factors that influence the formation and spatial distribution of surface water. This is ascribed to their directly controlling the distribution of relative negative topography, determining the characteristics of regional water flow, and redistributing water and heat. When elevation increases, the average annual precipitation increases, which is a "topographic enhancement" phenomenon derived from a combination of meteorological processes [34]. We selected the surface water indicated in 1988, 1993, 1998, 2004, 2010, and 2015 to analyze the combined spatial pattern of meteorological and elevation processes. These years were selected because they were abrupt transition years for the two types of surface water, as well as for precipitation and temperature.

With increasing elevation, the two types of surface water show a trend of extreme areal decline (Figure 10a,b), with 140 m elevation being the evident boundary. Most of the waterbodies are distributed at 14–140 m elevation and, particularly, at 60–80 m elevation, where the two types of water have the most extensive distribution area. Compared with the seasonal water, the permanent water showed a fluctuating decreasing trend in the elevation range 100–140 m. This finding is ascribed to nearly 100 large- and medium-sized reservoirs, such as the Nanwan, Banqiao, Boshan, Meishan, and Shishankou reservoirs, which are all located at an elevation range between 100 and 134 m. Similarly, there are nearly a thousand small reservoirs in the upper reaches of the major tributaries of the Huaihe River, such as the Huashan, Songjiachang, and Zhaozhuang reservoirs at elevations between 140 and 220 m. Consequently, fluctuations also occur in this elevation range. Seasonal water, potholes, rice fields, main canals, and the like are related closely with human activities. Therefore, as the population decreases at higher elevations and attendant human activities decrease, the seasonal water area also decreases sharply. Furthermore, as seasonal water is distributed widely in low-elevation areas, it is more susceptible to floods, such as regional floods caused by heavy precipitation in 1993 and 1998. This difference in area is particularly evident in plain and hilly areas below 160 m elevation.

In Figure 10c,d, the dynamic variable curve obtained from combining wet and dry years was higher than other years. The dynamic changes in the extent of permanent water during the wet and dry years are evidently greater than seasonal water. The region with a large dynamic change in the permanent water appears in the high-elevation region, whereas such change in the seasonal water varies with elevation, is not evident. This finding is attributed to water sources, such as reservoirs, potholes, and ponds in mountainous areas, with low precipitation and high surface evaporation in dry years. In addition, factors such as the supply of downstream irrigation, and ecological and domestic water use lead to part of the permanent waterbody being more vulnerable to the effects of wet or dry years, as well as being susceptible to change. Furthermore, because of the flood and drought prevention

measures of upstream reservoirs, the area occupied by seasonal water is controlled, which is one of the main reasons for the minimal interannual dynamic change in these waterbodies.

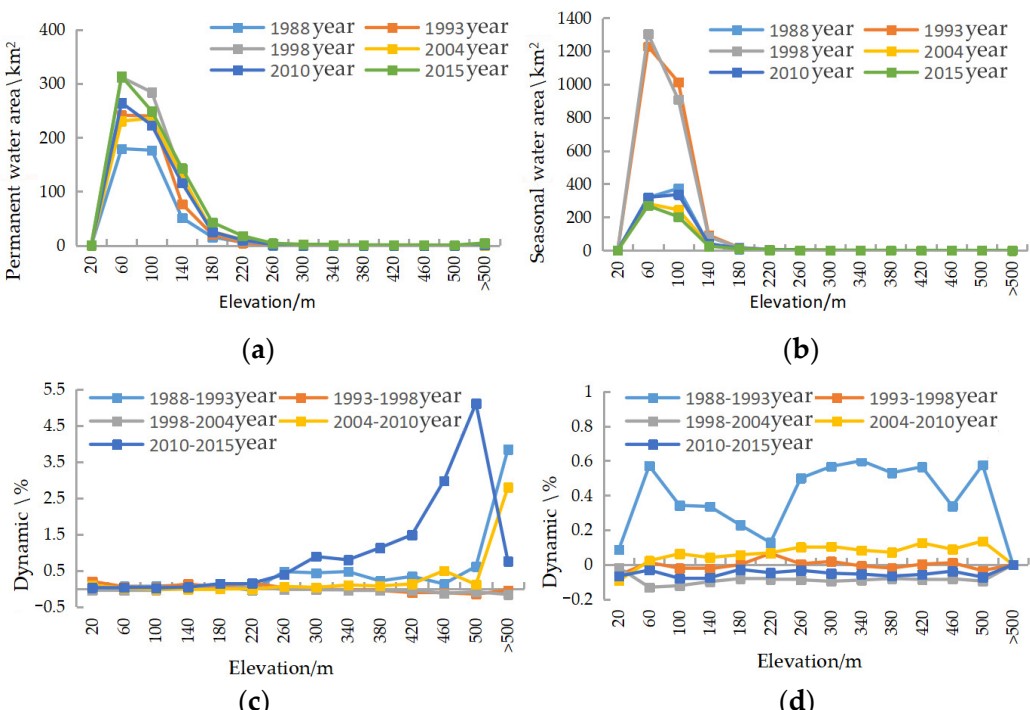

**Figure 10.** The two types Surface water area and dynamic changes at different elevations. (**a**) Permanent water area vs. elevation, (**b**) seasonal water area vs. elevation, (**c**) interannual dynamics of permanent water, and (**d**) interannual dynamics of seasonal water.

## 4. Discussion

### 4.1. Modeling Approaches

The construction of probabilistic statistical learning methods is based on data, and in view of the immense volume of available geospatial data, it is an effective approach to solve the perceived problem of being "data rich and knowledge poor" [35]. In addition, the frequency mapping rules that used all available "good" pixels rather than only high-quality images are also improving. This is attributed to the ability to make full use of all the acquired images, and to conduct land cover mapping with strong timelines and consistent spectral resolution. Although more scholars have considered this aspect, few theoretical models and processes of frequency mapping have been constructed, which is not conducive to the popularization and comparative application of frequency mapping technique. With the help of the Markov chain theory and the frequency ratio calculation model, we generated time-series remote sensing maps of surface water on an interannual scale. Compared with the JRC global surface water dataset, we found a clear difference in the seasonal water area. This finding could be attributed to the use of different data sets during mapping or could be related closely to the generation strategies. Ref. [9] used the monthly weighted method to generate the FR maps. Owing to the instability of the regional shallow-water culture, the difficulty of extracting constructed wetlands increases [36], which is not conducive to large-scale mapping in wetland systems mapping. Pekel et al. focus on global water extraction, so, their seasonal waterbodies tend to point to the beaches of rivers and lakes, neglecting mixed aquaculture areas with regional characteristics. With regard to the FR mapping in our study, we adopted image filtering to eliminate data with a total cloud cover of over 50% and pixels with cloud cover of over 50%. This was to obtain the top-layer reflectivity data of all the available sensors and to construct the cartographic dataset. However, the two sets of data were sufficiently consistent with regard to permanent

water areas, indicating the feasibility of frequency mapping of all Landsat 4–8 images by means of Markov chain theory [37].

### 4.2. Model Validation and Field Sampling

Generally, gield survey sampling or high-resolution image data indicate the true value of a water body, and the accuracy of the extraction results is determined based on several parameters [38]. This evaluation assesses single-phase extraction results for frequency mapping. As multiple-phase image sets are used for mapping over a certain period, conventional accuracy evaluation methods are inevitably not applicable. Compared to previously published land cover products, this product meets the accuracy of multitemporal mapping. Furthermore, the accuracy of the single-time product was found to mainly depend on the selection or distribution of samples [39]. In most cases, accuracy shows the validity of the ground object extraction model rather than the consistency between ground object classification and the ground truth. The validation dataset of the JRC data is derived from global random samples; therefore, its threshold division and accuracy evaluation system for permanent and seasonal water consider global water. Even within the same region, differences in spectral characteristics of surface water are caused by factors, such as the composition of the bottom material, depth, water quality, and influence of the surrounding environment. Additionally, the image features on the remote sensing images were unbalanced. Therefore, using a unified model to accurately extract surface water information is challenging [40]. A classifier is often used with the workflow of the classification algorithm to identify correct and representative training samples, calculate, and classify all pixels. However, because the limited training samples are lack of representativeness, the reoptimized algorithm is unable to improve classification accuracy [41]. The evaluation results of remote sensing interpretation products show that a region-specific land cover map is more accurate than the global land cover map [42–44].

### 4.3. Influence of Extreme Climate

As the interannual variation in air temperature is lower than that in precipitation, the surface water in the research area strongly correlated with precipitation at the interannual scale; however, the surface water did not evidently correlate with interannual temperature variation. The correlation between waterbodies area and precipitation substantially increased with rising temperature, particularly between seasonal water and precipitation, which showed a strong correlation over 29 years. Recently, extreme weather events, such as rainstorms, floods, droughts, and sunspot events [45], have become increasingly common in the study area. El Niño/La Niña events frequently result in heavy precipitation, especially during the flood season, which generates a concentration of precipitation in the area, increasing the contribution and frequency of extreme precipitation. Furthermore, heavy precipitation is usually followed by large periods of drought in the catchment, owing to the difference in the El Niño/La Niña occurrence cycle and water vapor transport mechanism. This phenomenon can be attributed to the typical airflow transport mechanism and landform, which also leads to lagging drainage of the surface water circulation system [46]. These factors weaken the interannual correlation between surface water area and precipitation. Therefore, upon analyzing the sequence diagram of climatic factors and the two types of surface waterbodies, we found that a temporal lag in correlation could exist between climatic factors and surface water. For example, in 2005 and 2007, regional floods were induced by heavy precipitation without any considerable increase in the surface waterbodies over the same period. However, in 2006 and 2008, when precipitation was low, the surface water area did not decrease with the lack of precipitation.

### 4.4. Influence of Elevation

The areal distribution and dynamic changes of the two types of water differed in each period and the water area decreased substantially as elevation increased. In years of extreme climate, the ratio of areal change in permanent water at different elevation

points was significantly higher than seasonal water. A dynamic change of permanent water occurred in areas >220 m elevation, where hundreds of large- and medium-sized reservoirs are widely distributed. The water supply pattern of these reservoirs has a substantial effect on the spatial distribution of seasonal water and form a trend effects in the south and north sub-catchments. This phenomenon is consistent with the results of Shi and Zhang [47], which indicates that the change in water resources in the runoff area of the plains and basins below mountain passes is controlled mainly by human water consumption. In contrast, the change in water resources in alpine runoff areas is affected mainly by climatic change. These findings imply that precipitation brought by meteorological processes and topographic enhancement will control the spatial distribution of permanent water. Meanwhile, human activities has substantially impact on the distribution of seasonal water. Above all has led to the spatial and temporal redistribution of surface water resources, changes in surface water structures and patterns, thereby causing changes in surface water cycle.

## 5. Conclusions

The main findings include:

1.  In this study, the pixel frequency calculation was optimized by introducing the Markov chain probability method in calculating the frequency of inter-annually classification of "water" and "non-water". The Markov chain probability method was based on the analysis of mixed border between stable waterbody and seasonal waterbody and criteria of Markov chain probability calculation;

2.  Over the past 30 years, eight abrupt changes in rainfall occurred in the study area, all consistent with the occurrence of El Niño/La Niña events. Owing to the influence of El Niño/La Niña events and climate warming, the spatial and temporal distribution of precipitation in the study area is uneven. The uneven distribution of surface water resources will inevitably result in the upward movement of key water conservancy projects;

3.  At an interannual scale, the correlation between surface water and precipitation is significantly higher than temperature. These two climatic factors show a strong correlation during wet and dry years. The correlation of seasonal water to both precipitation and temperature were all significantly higher than that of permanent water;

4.  Since 1993, a clear increasing trend in temperature has been observed, with a significant increase after 1999 because of global warming. Inter-annual variation of waterbodies and elevation showed that wet and dry years had fewer impact on seasonal water than permanent water.

**Author Contributions:** Conceptualization, Z.L. and D.C.; validation, H.W., F.Q. and D.C.; formal analysis, H.W., Z.L., J.Z. and D.C.; investigation, H.W., F.Q. and D.C.; formal analysis, H.W., Z.L., J.Z. and D.C.; resources, H.W. and F.Q.; data curation, H.W., Z.L., J.Z. and D.C.; writing—original draft preparation, H.W., Z.L., J.Z. and F.Q.; writing—review and editing, H.W. and F.Q.; visualization, H.W., Z.L. and J.Z.; and supervision H.W. and F.Q. All authors have read and agreed to the published version of the manuscript.

**Funding:** (2005DKA32300): Ministry of Education Base Major Project (No. 16JJD770019), Key Program of Henan Institute of Space-time Big Data Industry Technology (No. 2019DJA01), and School-level project of Hanshan Normal University (No. XN201920).

**Institutional Review Board Statement:** Not applicable.

**Informed Consent Statement:** Not applicable.

**Data Availability Statement:** The data presented in this study are available on request from the first and corresponding author. The data are not publicly available due to the thesis that is being prepared from these data.Finally, we acknowledge the two reviewers and editors of the special issue of the journal who provided very helpful comments that helped improved the final version of the manuscript.

**Conflicts of Interest:** The authors declare no conflict of interest. The sponsors had no role in the design, execution, interpretation, or writing of the study.

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
