# Peer review of "Spatio-Temporal Extraction of Surface Waterbody and Its Response of Extreme Climate along the Upper Huaihe River"

_sustainability, doi:10.3390/su14063223_

Round 1
Reviewer 1 Report
The paper was an interesting case study on the use of time-series cartographic technology and mutation tests, to investigate the effect of extreme events on rainfall, temperature, and surface water along the upper Huaihe River in China. It should be considered for publication after some significant revisions.
Abstract and Introduction:
Are fine, overall.
Materials and Methods
More information provided in the sub-section "Overview of the Study Area" without any reference?
Information that would have been very helpful to interpret the reletionship between rainfall and elevation is omitted altogether. This information include especially a map showing the general topography and hydrological netwrok of the study area so that the reader can assess the topographic influences on spatial distribution of rainfall.
- Figure 1: The legend is not visible. The resolution of the figure needs more care.
- Figure 3: In the flowchart, authors should be distinguish between the "water extraction model" in the two levels.
Materials and Methods
Figure 6 a and b: Please use the same legend of surface water classification. In 6a the authors used "permanent water, seasonal water" and in 6b "permanent, seasonal". Please update this legend.
In this section authors are invited to introduce how the "Accuracy Evaluation" is calculated. At least references should be provided.
Results
Section 3.2.1. reading the text we found 30 years ( 1987 to 2016) but in the figure 7 we found only 29 years (1987 to 2015). In the same figure the different dashed lines mean the precipitation level not the water body level (It is somewhat confusing). If yes, please in the legend use "low precipitation level, normal precipitation level and high precipitation level".
Section 3.2.2. All figures of this section need more care. Please upload figures with a high resolution.
How do you explain the low and negative correlation between raifall and seasonal waterbodies in 2000 (Figure 9a).
Comparing figure 6 and figure 10, I think there is somewhat confusing! looking at the figure 6 the surface water body is frequent at the upstream part (according to figure 6) and figure 10 shows a contradiction. Here the reader needs information about elevation of the study area to read and compare the results. Please use the logarithmic scale for Y-axis of figure 10a and 10b? This allow the better show the variation of surface water bodies areas!
The discussion needs more care.
Author Response
- Materials and Methods
Q1: More information provided in the sub-section "Overview of the Study Area" without any reference?
ANS:Add references to the section.
Q2: (1)Figure 1: The legend is not visible. The resolution of the figure needs more care;(2) Figure 3: In the flowchart, authors should be distinguish between the "water extraction model" in the two levels. (3)Figure 6 a and b: Please use the same legend of surface water classification. In 6a the authors used "permanent water, seasonal water" and in 6b "permanent, seasonal". Please update this legend.
ANS: (1)Figure 1 is re-redrawn with improved resolution. (2) Figure 3 has been normalized according to the recommended pattern for drawing a citation chart. (3) The legend annotation of Figures 6a and b was unified. Change it to “permanent water” and “seasonal water”.
Q3:In this section authors are invited to introduce how the "Accuracy Evaluation" is calculated. At least references should be provided.
ANS:The accuracy evaluation algorithms in this paper are commonly used models for remote sensing image interpretation, it is not necessary to explain the algorithms and the calculation process. The modification plan is to add the accuracy evaluation method references in this part.
2.Results
Q4:Section 3.2.1. reading the text we found 30 years ( 1987 to 2016) but in the figure 7 we found only 29 years (1987 to 2015). In the same figure the different dashed lines mean the precipitation level not the water body level (It is somewhat confusing). If yes, please in the legend use "low precipitation level, normal precipitation level and high precipitation level".
ANS: The description in this paragraph is based on the analysis of the annual surface water area and precipitation from 1987 to 2015,so the year 2016 is changed to 2015.According to the experts' suggestions, the three types of water body levels in Figure 7 are changed to "Low precipitation level, Normal precipitation level and high precipitation level", respectively.
Q5:Section 3.2.2. All figures of this section need more care. Please upload figures with a high resolution.
ANS: Rechecked figures in this paragraph with the context. The four images in Figure 8 are replaced with high-resolution images.
Q6:How do you explain the low and negative correlation between raifall and seasonal waterbodies in 2000 (Figure 9a).
ANS:It should be pointed out in particular that, the pre-flood season regional drought in 2000, “abrupt alteration from drought to flood” occurred before the plum rains season. Compared with other years with “abrupt alteration from drought to flood”, precipitation in this year was more plenty and concentrated. Following the alteration, there was basin-type flood. On the interannual scale, it showed more rainfall, but a smaller seasonal water for extended dry period. There was a negative correlation between the precipitation and seasonal water area.
The description is added in section 3.2.2.
Q7:(1) Comparing figure 6 and figure 10, I think there is somewhat confusing! looking at the figure 6 the surface water body is frequent at the upstream part (according to figure 6) and figure 10 shows a contradiction. Here the reader needs information about elevation of the study area to read and compare the results. (2) Please use the logarithmic scale for Y-axis of figure 10a and 10b? This allow the better show the variation of surface water bodies areas!
ANS:(1)Figure 10 shows statistics of the water area collected by elevation. The research area is in the upper reaches of the Huaihe River, with an elevation above 180m. Waterbodies in this area are mainly large-scale stable ones. Figure 6 clearly shows the distribution of these large waterbodies. The region of elevation below 180m is covered with medium and small reservoirs, ponds and small water bodies, mostly seasonal water bodies. For the scale of Figure 6, these small water bodies are not clearly drawn out. Therefore, the water area in the region of elevation below 180m (Figure 10) is larger. (2) In addition, Figures 10a and 10b show the distribution of the two types of waterbodies on different elevations. The figures only adopt raw data, without using the method of logarithmic scale for the Y-axis. Figures 10c and 10d are drawn and interpreted based on Figures 10a and 10b, for the convenience of reading and understanding.
Q8:The discussion needs more care.
ANS: Cut some paragraphs short. In particular, the language of section 4.2 、4.3 and 4.4 was re-edited.
During this round of revisions, I have checked word by word the place you mentioned, and measured my words so as to express my thoughts as more accurately and appropriately as possible. For other paragraphs, I have also checked line by line whether there are grammatical mistakes or inappropriate usage of words. I hope that you will find that this revised manuscript has been satisfactorily improved, and worthy of publication in sustainability. I am looking forward to hearing from you soon.
Thanks for your time and consideration. Best wishes.
Yours Sincerely,
- W, Ph.D.
Department of Geographic Sciences
Hanshan Normal University
Chaozhou, Guangdong 521041, China.
Tel: (86) 15913011102
Email: wanghang20001@163.com

Reviewer 2 Report
Surface waterbody evolution under the influence of extreme climate based on time-series cartographic technology
Dear Authors
The basic science of this paper has been conducted to a good and appropriate standard. The author and his team wrote this paper according to journal scope. I reviewed this paper and I found there is no novelty in this paper. The authors extract seasonal water surface bodies with models. Many authors also used these models in other research. Most important many flaws to write this paper. I don’t want to reject the paper because the author tries to write a paper. I hope the author will revise these sections according to my suggestions. I found some major, major revisions. All these are given below;
General Comments
- Check all citations. Citations do not reflect the study.
- Some sentences are very long in the whole manuscript. plz split
- I also found there are very big paragraphs. Revise some big paragraphs.
- Is this study from 1987-2018 or 1987-2016??????????
Major, Major comments according to section and subsection wise
Title
- Title is not according to the study. The author should revise because the author used time-series data (1987 to 2018/2016) and a case study area, Title reflects novelty but there is no novelty. It’s a case study. The author should write a case study in the title
- Spatio-temporal extraction of seasonal surface waterbody …………..
Abstract
- Abstract is very general just like the end paragraph of the introduction part.
- Abstract is very very small and does not reflect the study. Try to elaborate further in better ways.
- Novelty should be explained at the end of the Abstract.
- Author tries to write in a better way but still, there are some mistakes. I do not agree about the abstract part. The author should rewrite abstract with the following headings
- Context and Background
- Objectives
- Material and Methods
- Results/ findings
- Purpose/ Novelty
- Introduction
- Introduction part is also small and does not reflect the study.
- Very less cited articles and mentioned not proper cited articles and proper places
- Paragraphs are very big
- In first paragraph, Author can explain the background of waterbodies with old references. Why author used latest references here. All these are not suitable.
- Introduction part is very short and not according to study. I am going to suggests you three latest papers, kindly read and revise some part of introduction and also cite these papers.
- Somasundaram D, Zhang F, Ediriweera S, Wang S, Li J, Zhang B. Spatial and Temporal Changes in Surface Water Area of Sri Lanka over a 30-Year Period. Remote Sensing. 2020; 12(22):3701. https://doi.org/10.3390/rs12223701
- Characterization of the 2014 Indus River Flood Using Hydraulic Simulations and Satellite Images. doi: https://doi.org/10.3390/rs13112053
- Flash Flood Susceptibility Assessment and Zonation by integrating Analytic Hierarchy Process and Frequency Ratio model at Chitral region, Khyber PakhtunKhwa, Pakistan, https://doi.org/10.3390/w13121650
- Check the whole manuscript and split long sentences.
- Author should revise the introduction part and write the introduction part in three comprehensive paragraphs and all paragraphs should reflect your study and novelty
- In the last paragraph author should explain why do you want to conduct this study, what are main purposes, what type of audience read your paper and then explain the objective of this research.
- Material and Methods
2.1. Overview of the Study Area
- Please don’t start paragraph from as shown ….
- In the study area, the author should write about the manmade influence in the study area. How many people live and what is the geographical and socio-economic importance of this area.
- Author should write about topography and elevation of this study area
- Figure 1 does not reflect proper study. The author should revise this figure as per the overview of the study area.
- Author should use DEM and show an elevation in figure 1a
- Legends are not appropriate in Figure 1.
- Figure 1, export in high resolution
- I can’t read the legend and see clearly figure 1.
- Which river is there and secondly what is the source of the river. Is it a permanent river or a seasonal river, define the depth and bend of the river in this section.
- In figure 1 author should show the population settlements.
- Data Source and Processing Platform
- How you get this type of data
- Author should split this part into two parts, datasets, and processing of datasets.
- Author explained this part in little better ways
- Author should follow some steps from this paper to write this part in better ways.
- Somasundaram D, Zhang F, Ediriweera S, Wang S, Li J, Zhang B. Spatial and Temporal Changes in Surface Water Area of Sri Lanka over a 30-Year Period. Remote Sensing. 2020; 12(22):3701. https://doi.org/10.3390/rs12223701
- Probability Calculation
- Line 133. Citation is not proper according to the journal
- Why do you use the Markov chain model. Explain in more detail.
- Kindly check equation style
- Author should split paragraphs according to context.
- Author explained this part in little better ways
- Model Introduction
- What is a model introduction, the author should replace this heading with model validation
- Frequency ratio (Fr)à Frequency Ratio (FR)
-
- Surface Water body Map
- Is it a map or maps? Surface Water body Mapà Seasonal surface waterbody mapping or Mapping of the seasonal surface water body.
- Line 203-206: English is very poor. What is combined? Write English in proper ways.
- Subject, verb, and object
- Again paragraphs are very big with jerks words. It’s very difficult to understand and read your paper.
- Revise figure 3 and present it in better ways. All Figures are not according to the international and well-reputed journal.
- Revise caption of figure 3. Is it a technical route????????????????????
- Results
- Accuracy evaluation
- I agree with this part. Some sentences are long. The author should rewrite/split that sentences.
- Line 263 to 293: its big paragraph
- I could not find in methodology, how to calculate accuracy assessment
- Export figures in high resolutions
- I can’t see clearly results in figure 6, secondly, this figure does not show the Spatio-temporal changes of seasonal and permanent waterbodies
- Effects of Extreme Climate on Surface Water
- Extreme Precipitation, Drought, and El Niño/La Niña
- I see, in some parts author writes the results till 2018, and in some sections, author explains till 206. I am a little confused……..
- Line 318: results 1987-2016?????
- I can’t understand this part. The authors should revise this part. They used very poor English.
- Secondly paragraphs are very lengthy
- Discussion
- Paragraphs are very lengthy
- The author used very poor English
- It’s better if they used professional services or native speaker
- Agree about this part. The author adds some more details about data limitations.
- Need to check some typo errors.
- Conclusion
- Conclusion part is very lengthy but it’s well-structured. It's better if they revise and reduce some text.
In the end, I would like to say about your study. I believe you did a great job but we still need some improvement in your paper. There are many English grammar and typo errors. I hope you will modify it very soon and resubmit it again in this journal. I want to see your study in a better way according to journal criteria. I don’t want to reject your study because authors spend a huge time conducting this type of study.
I will just focus on my comments.
Best Regards
Author Response
1.Title
Q1:Title is not according to the study. The author should revise because the author used time-series data (1987 to 2018/2016) and a case study area, Title reflects novelty but there is no novelty. It’s a case study. The author should write a case study in the title Spatio-temporal extraction of seasonal surface waterbody …………..
ANS: According to the expert's suggestions, the title is changed to “Study on the relationship between extreme climate and surface waterbody using time-series cartographic technology for the upper reaches of Huaihe River”.
2.Abstract
Q2:Abstract is very general just like the end paragraph of the introduction part.Abstract is very very small and does not reflect the study. Try to elaborate further in better ways.Novelty should be explained at the end of the Abstract.Author tries to write in a better way but still, there are some mistakes. I do not agree about the abstract part. The author should rewrite abstract with the following headings:Context and Background—Objectives--Material and Methods--Results/ findings--Purpose/ Novelty.
ANS: According to the expert's suggestions, and referring to the journal's requirements for the abstract, the Abstract part was rewritten. In the revised manuscript, the Abstract part added the research background and research novelty, the lines for research object and research result were further simplified.
3.Introduction
Q3:SIntroduction part is also small and does not reflect the study.Very less cited articles and mentioned not proper cited articles and proper places.Paragraphs are very big.In first paragraph, Author can explain the background of waterbodies with old references. Why author used latest references here. All these are not suitable. Introduction part is very short and not according to study. I am going to suggests you three latest papers, kindly read and revise some part of introduction and also cite these papers.
- Somasundaram D, Zhang F, Ediriweera S, Wang S, Li J, Zhang B. Spatial and Temporal Changes in Surface Water Area of Sri Lanka over a 30-Year Period. Remote Sensing. 2020; 12(22):3701. https://doi.org/10.3390/rs12223701
- Characterization of the 2014 Indus River Flood Using Hydraulic Simulations and Satellite Images. doi: https://doi.org/10.3390/rs13112053
- Flash Flood Susceptibility Assessment and Zonation by integrating Analytic Hierarchy Process and Frequency Ratio model at Chitral region, Khyber PakhtunKhwa, Pakistan, https://doi.org/10.3390/w13121650
ANS:We read and learned the structure layout and writing skills of these three literatures carefully, and benefited a lot. According to the comments and the three recommended articles, we added the three recommended literature and other appropriate references to this section. At the same time, we revised the introduction to add the necessary explanation of surface water research in this study area, and also elaborated the existing problems in current research, summarized the research ideas and objectives at the end of the introduction.
Q4: Check the whole manuscript and split long sentences. Author should revise the introduction part and write the introduction part in three comprehensive paragraphs and all paragraphs should reflect your study and novelty .In the last paragraph author should explain why do you want to conduct this study, what are main purposes, what type of audience read your paper and then explain the objective of this research.
ANS: The introduction part was rewritten, and the expression of long sentences had been modified into short sentences as far as possible. Based on the expert’s advice, three paragraphs were used. The first paragraph introduced the sequence mapping technique background, the second paragraph mainly expounded the necessity of studying the response relationship between extreme weather and surface water, and the last paragraph added research ideas and research purposes.
4.Material and Methods
(1)Overview of the Study Area
Q5:Please don’t start paragraph from as shown ….In the study area, the author should write about the manmade influence in the study area. How many people live and what is the geographical and socio-economic importance of this area.Author should write about topography and elevation of this study area.
ANS: Problems about the expression at the beginning of paragraphs, long sentences, etc., have been solved according to the reviewer’s suggestions. In addition, because this study takes the physic-geographical environment of the study area as the object and involves no man-made influence, the population and economy of the study area have not been explained.
Q6: (1) Figure 1 does not reflect proper study. The author should revise this figure as per the overview of the study area. Author should use DEM and show an elevation in figure 1a Legends are not appropriate in Figure 1.Figure 1, export in high resolution.I can’t read the legend and see clearly figure 1. (2) Which river is there and secondly what is the source of the river. Is it a permanent river or a seasonal river, define the depth and bend of the river in this section. In figure 1 author should show the population settlements.
ANS:According to the reviewer’s suggestions, I have constructed DEM as the base map of the study area, and remapped Figure 1a and b, where the blue river represents the mainstream of the Huaihe River that runs through the study area from west to east. Figure 1a marks the source of the Huaihe River. Furthermore, the overview part introduces the Huaihe River as a perennial river, and gives brief explanations about the flow direction of its mainstream.
(2)Data Source and Processing Platform
Q7:How you get this type of data?Author should split this part into two parts, datasets, and processing of datasets.Author explained this part in little better ways.Author should follow some steps from this paper to write this part in better ways.
Flash Flood Susceptibility Assessment and Zonation by integrating Analytic Hierarchy Process and Frequency Ratio model at Chitral region, Khyber PakhtunKhwa, Pakistan, https://doi.org/10.3390/w13121650
ANS:The structure of this chapter has been normalized according to the recommended pattern, which is divided into three parts, introduction of the research area, data collection and water extraction.
(3)Probability Calculation
Q8:(1)Line 133. Citation is not proper according to the journal. (2)Why do you use the Markov chain model. Explain in more detail. Kindly check equation style.(3) Author should split paragraphs according to context.Author explained this part in little better ways.
ANS:(1)As required by the journal, I have modified the citation format. (2)The Bayesian network is a model for simulating causality in the process of reasoning, of which the Markov chain is an exception. The state of the system at time t expressed by the discrete-time stochastic process of the Markov chain coincides with the logical relationship of the Huaihe River Basin, which is supplied by rainfall as the major water source, and of which the waterbodies are divided into permanent or seasonal water. The Markov chain model probability theory is thus adopted as the theoretical basis for frequency mapping and surface water classification. This part is a bright spot of the paper, and a supplement to extraction and accurate classification of surface water bodies based on remote sensing big data, as well as an improvement to frequency mapping theory and process that have been scarcely mentioned in existing research.(3) Some sentences have been re-constructed.
(4)Model Introduction
Q9:What is a model introduction, the author should replace this heading with model validation Frequency ratio (Fr)à Frequency Ratio (FR).
ANS: Change "frequency ratio" and "frequency mapping" to "FR" in this article. At the same time, the language of section 2.3.2 , 2.3.3 and 2.3.4 was re-edited.
(5)Surface Water body Map
Q10:(1)Is it a map or maps? Surface Water body Mapà Seasonal surface waterbody mapping or Mapping of the seasonal surface water body. (2)Line 203-206: English is very poor. What is combined? Write English in proper ways. (3)Subject, verb, and object.Again paragraphs are very big with jerks words. It’s very difficult to understand and read your paper.
ANS:(1)When it comes to the phase 32 water frequency maps, the term "FR maps" is adopted. When it comes to water extraction and mapping, the term "surface water" is utilized. When it comes to classified extraction and mapping, the terms "seasonal water" and "permanent water" are used respectively ". (2) After a check, I found the term "combined" to appear twice in the original text. In the revised manuscript, Line176 was re-edited, and the term "combined" at Line207 was replaced by "refer to". (3) I have revised the long sentence and deleted the rarely-used terms such as "shrimp fields", which intends to express a hybrid model of aquaculture and rice planting. As such model has its unique regional characteristics and no professional terms of such model have yet been found, the model has been deleted in order to avoid ambiguity.
Q11:Revise figure 3 and present it in better ways. All Figures are not according to the international and well-reputed journal.Revise caption of figure 3. Is it a technical route?
ANS: Figure 3 has been normalized according to the recommended pattern for drawing a citation chart.
5.Results
(1)Accuracy evaluation
Q12:(1)I agree with this part. Some sentences are long. The author should rewrite/split that sentences.Line 263 to 293: its big paragraph.(2)I could not find in methodology, how to calculate accuracy assessment.Export figures in high resolutions.I can’t see clearly results in figure 6, (3) this figure does not show the Spatio-temporal changes of seasonal and permanent waterbodies.
ANS:(1)Sentences especially long sentences have been reconstructed. (2) As overall accuracy, Kappa coefficient, etc., are common methods for remote sensing images, they are only annotated at their first appearance in the first row of Section 2.3.4. (3) Figure 6b mainly shows the distribution of surface water bodies in the study area, expressed by JRC 2015 seasonal water and permanent water, while Figure 6a gives the distribution of surface water, expressed by the mapping method proposed in this paper. The purpose is to compare the two figures in the context of the same year, demonstrate the accuracy of the proposed frequency map of surface water bodies and provide a reliable data source for studying and analyzing the reaction of surface water bodies to extreme weather.Therefore, no mapping analysis of temporal and spatial changes of surface water herein.
(2)Effects of Extreme Climate on Surface Water
Q13:(1)I see, in some parts author writes the results till 2018, and in some sections, author explains till 206. I am a little confused……..Line 318: results 1987-2016?????I can’t understand this part. The authors should revise this part. They used very poor English. (2)Secondly paragraphs are very lengthy.
ANS:(1)The description in this paragraph is based on the analysis of the annual surface water area and precipitation from 1987 to 2015,so the year 2016 is changed to 2015.(2)Sentences especially long sentences have been reconstructed, and the language of section 3.2.1 was re-edited.
6.Discussion
Q14:(1)Paragraphs are very lengthy.The author used very poor English.It’s better if they used professional services or native speaker.Agree about this part. (2)The author adds some more details about data limitations.(3) Need to check some typo errors.
ANS:(1)The language of section 4.2 and 4.3 was re-edited. (2) Data limitations and algorithm limitations are described in detail in sections 2.3.4, 4.1, and 4.2.,On the basis of the original text, the revised version also adds the disscussion of the above two contents.(3) I have checked word by word the typo errors in this section, and measured my words so as to express my thoughts as more accurately and appropriately as possible. Sentences especially long sentenceshave been reconstructed, in particular, the language of section 4.2 and 4.3 was re-edited.
7.Conclusion
Q15:Conclusion part is very lengthy but it’s well-structured. It's better if they revise and reduce some text.In the end, I would like to say about your study. I believe you did a great job but we still need some improvement in your paper. There are many English grammar and typo errors. I hope you will modify it very soon and resubmit it again in this journal. I want to see your study in a better way according to journal criteria. I don’t want to reject your study because authors spend a huge time conducting this type of study.
ANS:This study relies greatly on field research, data collection and processing. Although I have tried my best to display the work I have done, I still feel sorry for your unpleasant reading experience due to my limited English competence (sometimes the wrong usage of words). During this round of revisions, I have checked word by word the place you mentioned, and measured my words so as to express my thoughts as more accurately and appropriately as possible. For other paragraphs, I have also checked line by line whether there are grammatical mistakes or inappropriate usage of words. Thank you again for your valuable comments!
I hope that you will find that this revised manuscript has been satisfactorily improved, and worthy of publication in sustainability. I am looking forward to hearing from you soon.
Thanks for your time and consideration. Best wishes.
Yours Sincerely,
- W, Ph.D.
Department of Geographic Sciences
Hanshan Normal University
Chaozhou, Guangdong 521041, China.
Round 2
Reviewer 2 Report
I reviewed this paper again. Author going to publish this paper in well reputed journal. How can I ignore author’s mistake in this manuscript? Last time I spend too much time and recommended major revision but I think author don’t want to publish this paper in this journal.
I matched revision with 1st submission. I found author just highlighted some lines. I did find any changes in highlighted version. (e.g. Line12, line 25-26 and many more). Now, I will recommend major revision. I hope author will check whole manuscript again and write proper response. Its last chance. Author should follow my all suggestions and comments one by one. I found some major revisions.
see document and write response of each point and author should follow my all suggestions and comments. I am not satisfy about your manuscript

Author Response
Manuscript ID: sustainability-1585058
3rd Mar, 2022
Dear Reviewer,
I am writing to submit a further revised manuscript entitled "Surface waterbody evolution under the influence of extreme climate based on time-series cartographic technology” by Wang et al. for publication as a full paper in sustainability. According to your email dated at 22th Feb, we have taken into consideration of the comments from you in preparing the second revised manuscript. The corresponding information were pointed out below, and all the changes are highlighted by yellow color in the “revised manuscript.doc”.
Q1:I found author just highlighted some lines. I did find any changes in highlighted version. (e.g. Line12, line 25-26 and many more).
ANS: In Line 12 of the revision, we’ve added "… and the effects remain largely unknown". Sentences highlighted in yellow have been partially modified, signifying lexical adjustments. e.g., "seasonal waterbody" into "seasonal water body", and "rainfall" into "precipitation". Revisions and editing made have been tracked and can be seen in sustainability-1585058.doc, which was submitted on February 18, 2022. I apologize for the misunderstanding. In the second revision draft, only the lexical changes were highlighted.
Q2:Check all citations. Citations does not reflect the study.
ANS: According to the suggestions, all citations were checked. In this revision, 7 new references were added and 10 were deleted, and the number of references was reduced from 50 to 47.
Q4: Is this study from 1987-2018 or 1987-2016?
ANS: Considering the timeliness of the paper, we conducted frequency mapping and accuracy evaluation for surface water from 1987 to 2018. However, precipitation and temperature data were only collected from 1987 to 2015, the relationship between surface water and extreme weather was studied from 1987 to 2015. If you think it was not suitable and need to unify the time period, we can make another drawing.
Q5: I am not satisfied about title of this study. I suggested but author ignore my
suggestions: Spatio-temporal extraction of seasonal surface waterbody
ANS: According to the suggestion, change the title of this paper to: Spatio-temporal extraction of surface waterbody and its response of extreme climate along the upper Huaihe River
Q5: Abstract is still general. Author did not follow my suggestions
Context and Background---Objectives---Material and Methods--Results/ findings
-- Purpose/ Novelty
ANS: According to the expert's suggestions, the Abstract part was rewritten. In the revised manuscript, added statements to the research background, research objectives and research novelty.
Q6: If you follow these papers, why you did not cite these papers in your
manuscript?
- Somasundaram D, Zhang F, Ediriweera S, Wang S, Li J, Zhang B. Spatial and
Temporal Changes in Surface Water Area of Sri Lanka over a 30-Year Period. Remote
Sensing. 2020; 12(22):3701. https://doi.org/10.3390/rs12223701
- Characterization of the 2014 Indus River Flood Using Hydraulic Simulations
and Satellite Images. doi: https://doi.org/10.3390/rs13112053
- Flash Flood Susceptibility Assessment and Zonation by integrating Analytic
Hierarchy Process and Frequency Ratio model at Chitral region, Khyber
PakhtunKhwa, Pakistan, https://doi.org/10.3390/w13121650
ANS: We double-checked the content of the references, removed inappropriate references.
- The first paper (https://doi.org/10.3390/rs12223701) was listed 8on the reference list for the revision and cited on Line 60-61.
- The second pater (https://doi.org/10.3390/rs13112053) was listed 6on the reference list for the revision and cited on Line 51.
- The third paper (https://doi.org/10.3390/w13121650) was the 19rdpaper on the revision’s reference list.
Q6: (1)Check whole manuscript and split long sentences.
(2)Author should revise introduction part and write introduction part in three
comprehensive paragraph and all paragraph should be reflect your study and novelty
ANS:(1)Check all the sentences longer than 2 lines, and modified them into short sentences as far as possible.
- According to the comments, the introductionpart has been revised and reviewed in three paragraphs. The first paragraph introduced the sequence mapping technique background, the second paragraph mainly expounded the necessity of studying the response relationship between extreme weather and surface water, and the last paragraph added research ideas and research purposes.
- Separate contents on research highlights have each been appended at the ends of the first and second paragraphs; some research details have also been added to the bottom parts of the second paragraph; lastly, descriptions of the research innovations have been attached to the third paragraph.
Q7: (1)Please don’t start paragraph from as shown ….
(2)In the study area, the author should write about the manmade influence in the
study area. How many people live and what is the geographical and socio-economic
importance of this area. (3)Author should write about topography and elevation of this study area
ANS:(1) Following the previous revisions made on all paragraphs led with the phrase “as shown”, secondary revisions have also been made.
(2)In the first paragraph of the 2.1 section, introductions to the river curvature, river source, river nature, and topography have been added.
(3) The second section has also been restructured to include a rundown of the population and economy in the research area.
Q8: (1) Figure 1 does not reflect proper study. Author should revise this figure as per overview of study area. Author should use DEM and show elevation in figure 1a ;Legends are not appropriate in Figure 1. ;Figure 1, export in high resolution
;I can’t read legend and see clearly figure 1. Which river is there and secondly what is source of river. Is it permanent river or seasonal river, define the depth and bend of river in this section. In figure 1 author should show the population settlements.
ANS:(1)Figure 1 has been modified as suggested, with Figure 1b mapped using a 30-meter DEM. Legends have also been added to both Figure 1a and Figure 1b. Figure 1b has been enlarged to increase the mapping resolution, Tongbai Mountain has also been added therein as a river source, while information detailing the locations of city, district and county level governments have also been specified.
Q8: 2.2. Data Source and Processing Platform
(1) How you get this type of data (data source?)
(2) Author should split this part in two parts, datasets and processing of datasets.
ANS:(1) Explanations on data sources and preprocessing had been previously laid out in 2.2.2 and 2.2.3 during the revision. All image data used in 2.2.1 were all acquired from GEE, this has been described in the second revision.
(2)The 2. section having previously been modified according to the first reference paper (https://doi.org/10.3390/rs12223701). This revision split the 2. section into three parts, study area, datasets and data processing.
(3)In addition,The contents of sections 2.3.1 and 2.3.2 were reintegrated into section 2.3.1 of the revised manuscript.
Thank you very much for your valuable comments! I hope that you will find that this revised manuscript has been satisfactorily improved, and worthy of publication in sustainability. I am looking forward to hearing from you soon.
Thanks again for your time and consideration. Best wishes.
Yours Sincerely,
- W, Ph.D.
Department of Geographic Sciences
Hanshan Normal University
Chaozhou, Guangdong 521041, China.

Round 3
Reviewer 2 Report
Dear Author
Thank you for your patience. I am glad to see your revised version paper. I hope you ll publish more papers in the near future.
Best Regard